

# How does humidity affect lidar-derived aerosol optical properties, and how do they compare with CAMS?

Frédéric Laly[1,2], Patrick Chazette[1], Julien Totems[1], Vincent Crenn[2], David Ledur[2], Alexandre Marpillat[2]

[1]LSCE/IPSL, CNRS-CEA-UVSQ, University Paris–Saclay, CEA Saclay, Gif sur Yvette, France
[2]ADDAIR Company, 78530, Buc, France

*Correspondence to*: Frédéric Laly (Frederic.laly@lsce.ipsl.fr)

**Abstract.**

From May to August 2020 and during summer 2024, aerosol backscatter and relative humidity profiles were measured in the
Paris region using the Water Vapour and Aerosol Lidar (WALI). The campaigns included observations on the Saclay plateau (48°42'42'' N / 2°8'52'' E) and in Paris (48°50'12'' N / 2°20'10'') during the 2024 Olympic Games. The high vertical (15 m) and temporal (15 min) resolution of WALI allow study of aerosol optical properties and water vapour under stable atmospheric conditions. This study focuses on characterizing aerosol hygroscopic growth using lidar derived backscatter coefficients as a function of relative humidity. Eight case studies were selected where the potential temperature gradient was
neutral and the water vapor mixing ratio was constant with height. These include long range pollution transport from the Benelux region, low hygroscopic aerosol events, and a sea salt-pollution mixture episode. Hygroscopicity was assessed using CAMS model analyses and reanalyses, allowing attribution of chemical composition to observed optical changes. Good agreement was found between lidar-derived hygroscopic properties and CAMS-inferred aerosol types. Lidar growth factors ranged from 0.3 to 1.5, with higher values linked to sea salt presence, consistent with literature values. Aerosols hygroscopicity
is also studied using Mie's theory, as aerosols can be considered nearly spherical. Differences between extinction and backscatter-based growth retrievals are interpreted through Hänel's formalism. The results demonstrate the capability of Raman lidar to constrain aerosol hygroscopicity, offering valuable input to chemistry-transport models and helping to reduce uncertainties in climate projections related to aerosol-cloud interactions.

## 1   Introduction

The interaction between atmospheric aerosols and water vapour is essential to consider when studying Earth's climate (IPCC, 2023). While water vapour is the most potent greenhouse gas, atmospheric aerosols also influence the energy balance of the atmosphere by more generally cooling the Earth surface (IPCC, 2023). Aerosols directly absorb or scatter the incident electromagnetic radiation from the Sun and the Earth/Atmosphere system (Raut and Chazette, 2008a; Thorsen et al., 2020), which is known as the direct effect. These interactions can also indirectly modify the thermodynamic parameters of the
atmosphere by the semi-direct effect (Ramanathan et al., 2001; Hansen et al., 1997; Koren et al., 2004) or alter cloud properties, impacting both cloud albedo and lifetime (the indirect effects; Albrecht, 1989; Seinfeld et al., 2016; Twomey, 1977). These effects are strongly dependent on the chemical composition of aerosols, their optical properties, their lifetime, and the prevailing atmospheric and synoptic conditions (Tombette et al., 2008). All of these factors make it difficult to correctly assess the radiative impact of aerosols on the Earth's energy balance, resulting in significant uncertainties (IPCC, 2023).

Water vapour plays a fundamental role in cloud formation through interaction with cloud condensation nuclei (CCN) (Novakov and Penner, 1993) but also on aerosol ageing processes, which significantly modify their radiative properties. Because of the heterogeneity of aerosol spatial distribution and the diversity of their sources, which drive their chemical composition, interactions between aerosols and water vapour are always difficult to assess. The inaccurate estimation of water vapour –



aerosol interactions thus represents an uncertainty of 30 % in the radiative forcing of aerosols on the climate (Haywood and Boucher, 2000). Indeed, aerosols can have a hygroscopic capacity, which corresponds to their ability to take up water vapour. This property was first described theoretically by Hanël (Hänel, 1976), who showed that an aerosol increases in size and decrease in complex refractive index as humidity increases, thereby modifying its optical and physicochemical properties.

Although considerable progress has been made in atmospheric transport models, accurately accounting for the hygroscopic behaviour of aerosols remains particularly challenging (Wang et al., 2013). This difficulty is partly due to the available measurement techniques, which remain limited in the tropospheric column. Techniques such as humidified tandem nephelometers (Covert et al., 1972) enable measurements under controlled humidity, but only at a specific point, which requires airborne means to access the atmospheric column. In addition, these measurements are not made directly in the atmosphere, but in a controlled humidity chamber, which may introduce bias due to the limitations of such an approach. In several studies, in situ measurements, such as humified nephelometers, were also used to determine the hygroscopic potential of aerosols, confirming Hänel's theory (Randriamiarisoa et al., 2006; Raut and Chazette, 2008b; Titos et al., 2016) during moisture growth. However, these approaches require an assessment of the changes in the physicochemical properties of the aerosols over several temporal evolutions of humidity conditions.

In response to these limitations, lidar measurement is the most accurate method of characterising aerosol hygroscopicity, as it avoids disturbing the sampled environment. Nowadays, Raman lidar instruments allow to characterise the hygroscopicity of aerosols by simultaneously measuring of the relative humidity (RH), via water vapour and temperature channels (Behrendt, 2006; Chazette et al., 2014b; Whiteman et al., 1992), and backscattering coefficient. In several studies, lidars were used to complete the characterisation of the hygroscopic power of aerosols, by studying changes in the extinction coefficient of aerosols according to RH (Dawson et al., 2020; Veselovskii et al., 2009) or backscatter coefficient (Chen et al., 2019; Feingold and Morley, 2003; Granados-Muñoz et al., 2015; Haarig et al., 2017; Lv et al., 2017; Miri et al., 2024; Navas-Guzmán et al., 2019; Perez-Ramirez et al., 2021; Sicard et al., 2022). These studies show that lidars can be used to study the growth of aerosols due to hygroscopic effects, and to identify the types of aerosol present. Lidar measurements offer the ability to characterise the hygroscopicity of aerosols under real atmospheric conditions, even when RH is close to saturation, without altering the state of the particles. This approach does assume that the air mass being studied is in mixing equilibrium and that the aerosols do not vary in number or dry chemical composition, whereas RH is the only varying parameter in the air column. Therefore, it is necessary to identify these specific situations in the lidar profiles and verify the assumptions made afterwards.

The study presented in this article is based on lidar measurements conducted in the Paris Basin using the Water Vapour and Aerosol Lidar (WALI) (Chazette et al., 2014b; Totems et al., 2021), deployed at the Laboratoire des Sciences du Climat et de l'Environnement (LSCE) at CEA Saclay (48°42'42'' N / 2°8'52'' E) in 2020 and at the Observatoire de Paris (48°50'12'' N/ 2°20'10'') in 2024. The field campaigns took place in May–August 2020 and in July–August 2024. These campaigns allowed the collection of lidar profiles relevant to study the effect of humidity on the retrieval of aerosol optical properties. The hygroscopic behaviour of aerosols during different periods of air mass advection is analysed, taking the origins of the air masses and the aerosol chemical composition into account. The Copernicus Atmospheric Monitoring Service (CAMS) aerosol speciation reanalyses and analyses are studied in relation to the lidar measurements. In order to study the origin of air masses, we use the Hybrid Single Particle Lagrangian Integrated Trajectory (HYSPLIT) model.

Section 2 presents the instrumental setup and the modelling data. The WALI instrument and the CAMS products are introduced. The lidar equations and the methodology for studying aerosol hygroscopicity are presented in Section 3. Section 4 presents the different case studies with the associated uncertainties. Section 5 shows how, using Mie modelling, it is possible to relate the different hygroscopicity parameters as defined by Hänel (Hänel, 1976), thereby demonstrating the effectiveness of the method. The consistency with the CAMS analyses and reanalyses is given in Section 6, followed by the conclusion.



## 2 Instrumental setup and modelling

### 2.1 Site location

The Water Vapour and Aerosols Lidar (WALI) (Chazette et al., 2014; Totems et al., 2021) was deployed from May to August 2020 at the Laboratoire des Sciences du Climat et de l'Environnement (LSCE, 48°42'42'' N / 2°8'52'' E) on the Commissariat à l'Energie Atomique et aux énergies alternatives (CEA) campus in Saclay. This location is marked by a red cross in Figure 1.

The LSCE is approximately 14 km at the southeast of the Trappes Meteo–France station (represented by a blue cross in Figure 1), which provides temperature and humidity vertical profiles twice a day following the World Meteorological Organization (WMO) requirements. These measurements were used to calibrate the WALI lidar for retrieving atmospheric temperature and water vapour (Chazette et al., 2025). WALI was also deployed at the Observatoire de Paris (48°50'12'' N/ 2°20'10'') in July and August 2024.

The meteorology of the Paris Basin is largely governed by the dynamics of low and high-pressure systems over the North Atlantic (Freutel et al., 2013; Lemonsu and Masson, 2002; Massei et al., 2007). The prevailing winds usually come from the west, bringing humid maritime air masses, or from the northeast, bringing more continental and often more polluted air masses after passing through the Benelux region, the Ruhr area, or the North Sea. The latter situations mainly occur during spring (Chazette and Royer, 2017) and sometimes during summer (Chazette et al., 2017). When high-pressure systems are positioned further to the east, air masses from northern Europe can be brought in via Germany, contributing to the arrival of air masses influenced by human activity. On the other hand, southerly winds are relatively scarce, and thus play a more limited role in influencing the transport of air masses toward the Paris area. Saclay and the Observatoire de Paris are therefore strategic sites for observing a wide range of atmospheric conditions and diverse origins of aerosols.

The period from April to June 2020 coincides with the establishment of anticyclonic conditions over the British Isles and France. This synoptic pattern favours stable atmospheric conditions and can enhance the long-range transport of air masses. During this season, emissions from major industrial regions such as the Ruhr and the Benelux are particularly significant, while agricultural and livestock activities also resume in northern France, contributing to the emission of various pollutants. These aerosol sources influence the composition of air masses circulating through the Paris Basin. It is worth noting that the observation period from May to August 2020 followed the initial lockdown due to the Coronavirus pandemic. As a result, this year was marked by an atypical pollution context, with atmospheric pollutant concentrations likely lower than climatological averages due to reduced industrial, traffic, and economic activities. Between June and August, the Paris Basin typically experiences summer meteorological conditions characterized by increased temperatures, episodes of atmospheric stagnation, and more frequent convective activity. While anticyclonic conditions can persist, Atlantic disturbances and occasional southerly flows can occur as observed in August 2024.

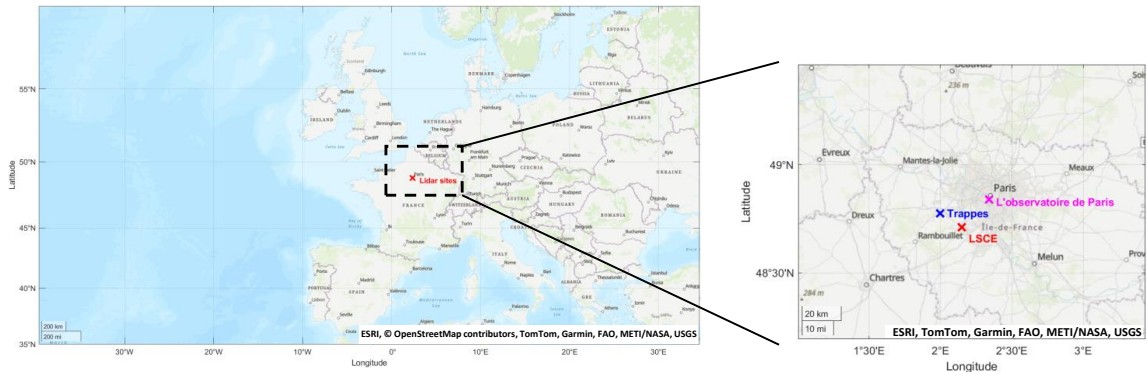

**Figure 1. Map showing the locations of the LSCE site in Saclay (red cross), the Observatoire de Paris site in Paris (magenta cross), and the Meteo-France radiosonde station in Trappes (blue cross). The map on the right shows a closer view of the Parisian Basin.**



## 2.2 Lidar instrument presentation

WALI is embedded in the Mobile Atmospheric Station (MAS) (e.g. Raut and Chazette, 2009). The optical architecture of the
115 lidar and its measurement performance for studying aerosols, water vapour and temperature are described in Totems et al.
(2021) and Laly et al. (2024). Pictures of the MAS truck and the lidar are given in Figure 2.

The lidar system uses as emitter pulsed frequency–tripled Nd:YAG laser manufactured by Lumibird Quantel, with an emission
wavelength of 354.7 nm. The laser beam is expanded using a beam expander (expansion factor x10), which allow to the meet
eye safety standards (EN 60825–1) at the chimney exit. The pulse energy is 100 mJ, while the pulse repetition rate is 20 Hz.
The reception system for Raman channels is a 150 mm Newtonian telescope, feeding filter–based spectral analysers, called
polychromators, via an optical fibre. This reception configuration allows full overlap (between the transmitter beam and the
receiver field of view), to be achieved at ~200 m above the lidar. For aerosol channels, the receiving system is a refracting
telescope (refractor) with lenses and polarising plate beamsplitters that separate the parallel and perpendicular polarisations
with respect to the emission, thus enabling, in addition to the optical properties of aerosols, the speciation of aerosols via their
depolarisation. Full overlap for aerosol channels is ~250 m. We use PXI (PCI eXtensions for Instrumentation) technology for
the acquisition system. It incorporates 12–bit PXI-5124 digitizers manufactured by NI® (https://www.ni.com/, last access
05/05/2025). The digitizers operate at a speed of 200 MHz, allowing for post–digitization photon counting. The temporal and
vertical resolutions, as well as the ranges of the lidar measurements, are given in Table 1. The lidar profiles used in this article
have a temporal resolution of 15 min and a vertical resolution of 15 m. It should be noted that the range of the lidar for
measuring water vapour and temperature is much shorter during the day than at night due to the sky background, which limits
the range of the Raman signals during daytime.

Note that radiosondes from the Meteo-France station at Trappes (48°46'26'' N / 2°0'37'' E) were used to calibrate the lidar
for water vapour and temperature as in Chazette et al. (2025) and Baron et al. (2022).

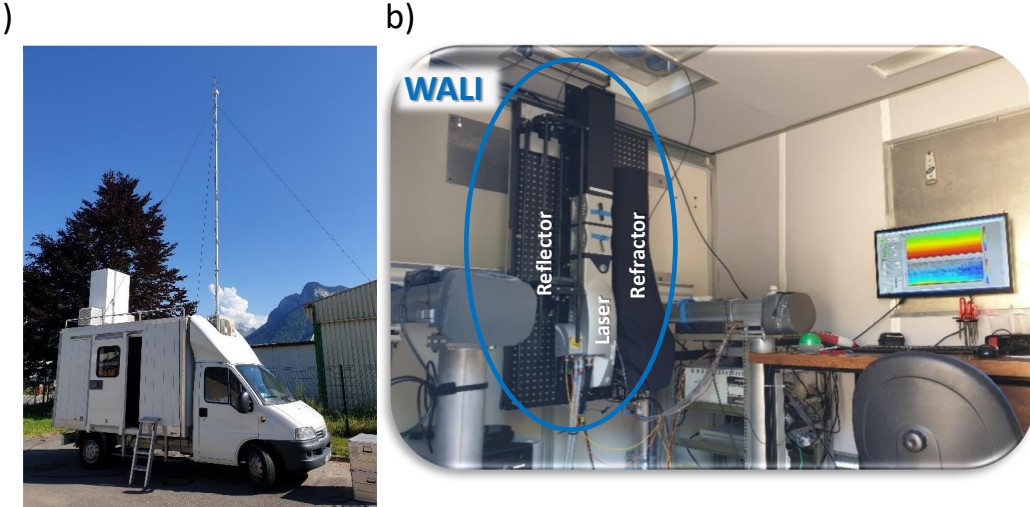

**Figure 2. Pictures of a) the MAS truck station transporting the WALI lidar; b) the WALI lidar on its optical table. The laser is at
the centre of the table. On the left is the reflector, a Newtonian telescope that enables Raman measurements. The refractor for cross-
polarised measurements of the lidar signal, is on the right. The lidar electronics are mounted in the rack to the right of the lidar.**

## 2.3 Modelling data

Different models are used to complement lidar measurements when analysing case studies, as well as to combine lidar
measurements with their outputs. Table 1 also summarises the spatiotemporal properties of the models.





### 2.3.1 CAMS analysis and reanalysis

To monitor atmospheric aerosol species concentrations, we use the Copernicus atmospheric monitoring service CAMS (Inness et al., 2019) global atmospheric composition forecasts (https://ads.atmosphere.copernicus.eu/datasets/cams-global-atmospheric-composition-forecasts/, last access 05/05/2025). The forecasts consist of more than 50 chemical species (e.g. ozone, nitrogen dioxide, carbon monoxide) and seven different chemical compounds of aerosols (mineral, sea salt, organic matter, black carbon, sulphate ions, nitrate ions and ammonium ions). The initial conditions of each forecast are obtained by combining a previous forecast with current satellite observations through a process called data assimilation. This is the best estimate of the state of the atmosphere at the initial forecast time step. This estimate is called the "analysis" and provides a globally complete and consistent dataset. This enables estimates in areas with limited observational data, or for atmospheric pollutants for which no direct observations are available. The forecast itself uses a model of the atmosphere based on the laws of physics and chemistry to determine the evolution of the concentrations of all species over time for the next five days. In addition to the required initial state, it uses either inventory- or observation-based emission estimates as a boundary condition at the surface. The analyses have the advantage of a better resolution than the reanalyses (0.4°x0.4° compared with 0.75°x0.75°). However, although the Copernicus website states that analyses are provided at a resolution of 3 hours, the resolution is reduced to 12 hours for downloaded data, whereas it remains at 3 hours for reanalysis. The reanalyses offer the advantage of greater accuracy because more resources are available for estimating the various atmospheric parameters. As discussed in Section 4, the two products complement each other by providing different useful parameters for this study. They were thus integrated to provide a unified perspective.

### 2.3.2 ERA5

To analyse the meteorological context of the various case studies presented in the article, the European Centre for Medium-Range Weather Forecasts (ECMWF) ERA5 reanalyses (Hersbach et al., 2020) have been used. ERA5 provides hourly data on a 31 km grid with up to 137 vertical pressure levels (https://cds.climate.copernicus.eu/datasets/reanalysis-era5-pressure-levels/, last access 05/06/2025). This detailed information covers the atmosphere from the surface to very high altitudes (1 hPa ~ 48 km a.m.s.l.). To check the meteorological contexts, we used the standard pressure level of 850 hPa (~1.5 km a.m.s.l.).

### 2.3.3 Lagrangian modelling (HYSPLIT)

The Hybrid Single Particle Lagrangian Integrated Trajectory (HYSPLIT; Stein et al., 2015; https://www.ready.noaa.gov/HYSPLIT_traj.php, last access 05/06/2025) model is used to identify the origins of the air masses in the case studies. HYSPLIT is a complete system for computing simple air parcel trajectories, as well as complex transport, dispersion, chemical transformation, and deposition simulations. The model calculation method is a hybrid between the Lagrangian approach, using a moving frame of reference for the advection and diffusion calculations as the trajectories or air parcels move from their initial location, and the Eulerian methodology, which uses a fixed three-dimensional grid as a frame of reference to compute pollutant air concentrations. HYSPLIT provides the probability density of the origin of air masses with a time resolution of 1 hour on a 0.25°x0.25° grid when the model is initialized using the wind fields of the Global Forecast System (GFS) (http://www.ncep.noaa.gov/, last access: 6 June 2025). The origin of air masses can be traced back over several days. In this study, the ensemble mode has been used with 27 five-day back trajectories.

**Table 1. Spatiotemporal resolutions of the WALI lidar and numerical models**

|  | WALI | CAMS | ERA5 | HYSPLIT |
|---|---|---|---|---|
| Data type | Observation | Analysis / Renalysis | Reanalysis | Lagrangian retrotrajectory |





| | | | | |
|---|---|---|---|---|
| Horizontal resolution | _ | 0.4°x0.4° / 0.75°x0.75° | 0.25°x0.25° | 0.25°x0.25° |
| Vertical resolution / Pressure level resolution | 15 m | 25 – 100 hPa | - | - |
| Altitude/Level considered | 0 - 3 km | 0 – 4.5 km | 850 hPa (~1.5 km) | - |
| Temporal resolution | Native: 1 min Operational : 15 min | 12 hours / 3 hours | 1 hour | 1 hour |
| Altitude range (a.g.l.) Raman channels | 0 – 2 km (Daytime) 0 – 10 km (Nighttime) | - | - | - |
| Elastic channels | 0 – 10 km | | | |

## 3 Theory to retrieve lidar products

This section describes the equations used to retrieve water vapour, temperature and aerosol optical properties from lidar measurements.

### 3.1 Lidar-derived-water vapour mixing ratio

Vibrational Raman lidars acquire signals corresponding to the backscattering of dinitrogen (channel N) and water vapor (channel H) molecules from the atmosphere. With laser emission at a wavelength of 355 nm, Raman detection of nitrogen and water vapour is performed at ~387 and 407 nm, respectively (first Stockes vibrational lines). During the acquisition process, the lidar profiles are sampled with a raw vertical resolution of 0.75 m, which is downgraded to 15 m to improve the signal-to-noise ratio. A temporal averaging of 1000 profiles translates to approximately one recording every 50 s over the campaigns. The vertical profile of WVMR ($r_H$) is given against the altitude $z$ by:

$$r_H(z) = \frac{m_H(z)}{m_a(z)} \tag{1}$$

where $m_H$ is the mass of water vapor and $m_a$ the mass of dry air. This can also be written as follows:

$$r_H(z) = \frac{N_H(z)}{N_N(z)} \cdot \frac{M_H}{M_N} \cdot r_N \tag{2}$$

where $N_H$ ($N_N$) is the density profile for water vapour (nitrogen) and $M_H$ ($M_N$) the associated molar mass. The dinitrogen mixing ratio is $r_N$.

The WVMR lidar profiles is derived from the two channels H and N according to the relationship:





$$r_H(z) = K_0 \cdot \frac{O_N(z)}{O_H(z)} \cdot \frac{\langle S_H(z)/g_H \rangle_M}{\langle S_N(z)/g_N \rangle_M} \cdot C_m(z) \cdot C_a(z) \tag{3}$$

where $K_0$ is the calibration coefficient, determined by comparison with a reference source. In this case, the reference source is radiosondes. The overlap factor of channel $i$ is $O_i$. The variables $C_m$ and $C_a$ are the correction terms associated with atmospheric transmission due to molecules and aerosols, respectively. Their expressions are given in Chazette et al. (2014a) and Laly et al. (2024). The water vapor ($S_H$) and dinitrogen ($S_N$) channels are corrected for the detection gains $g_H$ and $g_N$ (e.g. Chazette et al., 2025), respectively. The WVMR is calculated on a time–average ($\langle \ \rangle$) of $M$ profiles.

### 3.2 Lidar-derived temperature

Rotational Raman lidar measures temperature by means of two distinct wavelengths of 354.1 nm (channel RR1) and 353.2 nm (channel RR2) on the anti-Stokes side of the rotational Raman spectrum of both dioxygen and dinitrogen (Totems et al., 2021; Baron et al., 2022). Firstly, the ratio $Q$, which depends on the atmospheric temperature ($T$), is defined as follows:

$$Q(z, T) = \frac{\langle S_{RR2}(\lambda_l, z, T) \rangle_M}{\langle S_{RR1}(\lambda_l, z, T) \rangle_M} \tag{4}$$

$Q$ is linked to $T$ by the empirical relationship (Behrendt, 2006):

$$Q(T) = \exp\left(\frac{a}{T^2} + \frac{b}{T} + c\right) \tag{5}$$

The expression of $T$ can then be derived as:

$$T = \frac{-2a}{b + \sqrt{b^2 - 4a(c - \ln(Q))}} \tag{6}$$

### 3.3 Lidar-derived relative humidity

Relative humidity $RH$ is derived as a function of atmospheric pressure, temperature and WVMR, using standard empirical relationships for the water vapor saturation pressure $P_{wv,sat}$. Here, we use the Arden Buck equation (Buck, 1981), which is sufficiently accurate between −40°C and +100°C and given by an exponential relationship:

$$P_{wv,sat} = 6.1121 . e^{\left(\frac{T}{T+257.14°C}\right).\left(18.678 - \frac{T}{T+234.5°C}\right)} \tag{7}$$

$RH$ can be calculated using the following equation:

$$RH = \frac{P}{P_{wv,sat}} \cdot \frac{r_{H_2O}}{r_{H_2O} + 621.991} \tag{8}$$

where $P$ is the atmospheric pressure in hPa, $T$ the atmospheric temperature in °C, and $r_{H_2O}$ the WVMR in g kg⁻¹. Pressure data are taken from ERA5 reanalysis for the period.

### 3.4 Lidar-derived aerosol backscatter coefficient

The aerosol backscatter coefficient $\beta_a$ is calculated using both the elastic lidar signal ($S_E$) and N₂–Raman signal ($S_N$). Detailed expression of $S_E$ is given in Chazette et al. (2014b). The two signals at the wavelengths $\lambda_E$ and $\lambda_N$ of the lidar channels are given at altitude $z$ by:

$$S_E(z) = K_E . (\beta_a(\lambda_E, z) + \beta_m(\lambda_E, z)) . T_m(\lambda_E, z)^2 . T_a(\lambda_E, z)^2 \tag{9}$$

$$S_N(z) = K_N . \beta_m(\lambda_N, z) . T_m(\lambda_E, z) . T_a(\lambda_E, z) \cdot T_m(\lambda_N, z) . T_a(\lambda_N, z) \tag{10}$$

for the elastic and N₂–Raman channel respectively. The subscript $a$ ($m$) stands for aerosols (molecules), $\beta_a$ ($\beta_m$) and $T_a$ ($T_m$) are the backscatter coefficients and the atmospheric transmissions. The system constants are $K_E$ and $K_N$ for the elastic and N₂–Raman channels, respectively.





Assuming that the spectral dependence of aerosol transmission remains constant in the lower troposphere, $\beta_a$ can be expressed as a function of the Ångström exponent $A$:

$$\beta_a(\lambda_E, z) = \left( \frac{S_E(z)}{S_N(z).K} \cdot T_m(\lambda_E, z)^{-0.29} \cdot T_a(\lambda_E, z)^{(A-1)} - 1 \right). \beta_m(\lambda_E, z) \tag{11}$$

with $K$ equal to the ratio of the system constants of the elastic and Raman channels ($K = K_E/K_N$). The $K$ parameter is calculated by considering an altitude where the aerosol contribution is negligible, typically between 5 and 6 km in altitude for

all the case studies. Note that the molecular contribution is calculated using the ERA5 data set and $A$ is derived from the AERONET data set for the Palaiseau station (https://aeronet.gsfc.nasa.gov/, last access 11 June 2025). The AERONET data for each case gives an Angstrom coefficient of around $1.4 - 1.5$, characteristic of pollution particles.

It is worth noting that WALI can also measure the particulate depolarisation ratio (PDR). The method to calculate this parameter is described in Chazette et al. (2014b). At the altitudes considered in our cases, we are dealing with pollution-type

particles that depolarise the signal very little (low PDR). PDR information is useful because it allows us to rule out the presence of terrigenous aerosols with strong depolarising properties.

### 3.5    How to highlight hygroscopicity from lidar measurements?

Reliable retrieval of aerosol hygroscopicity from lidar measurements requires specific atmospheric conditions to ensure that observed changes in optical properties are due to humidity effects alone. In particular, the studied layer must contain aerosols

of uniform composition and size distribution, while allowing for a temperature gradient with altitude — conditions typically found in a well-mixed layer. This leads to practical criteria commonly applied in lidar-based studies (Granados-Muñoz et al., 2015; Miri et al., 2024; Navas-Guzmán et al., 2019), which are: i) quasi-constant $WVMR$ ($r_H$) in the layer, ii) low and constant potential temperature gradient $d\theta/dz$. RH thus varies mainly because of the natural gradient of temperature with altitude. These atmospheric conditions mean that the increase in RH is the main cause of growth in atmospheric aerosols and changes in their

associated optical properties.

Using in situ airborne measurements over the Paris area, it has also been shown that, in an atmospheric layer where $\theta$ and $r_H$ hardly vary, the granulometry of aerosols remains almost the same within the layer (Chazette et al., 2005), once again guaranteeing that humidity alone is responsible for the change in the aerosol optical properties. Note that the case studies analysed in this work occur in the early evening, when a residual layer typically forms between 700 and 1700 m above mean

sea level (a.m.s.l.). During this period, air masses caught between the top of the residual layer and the free troposphere are characterized by a weak potential temperature gradient, indicating stable stratification and limited vertical mixing.

The hygroscopic properties of aerosols were first studied and theorised by Hänel (1976) and used by several authors (e.g. Boucher and Anderson, 1995; Chen et al., 2019; Gassó et al., 2000; Kotchenruther et al., 1999; Tang, 1996). These studies show that hydrophilic aerosols exhibit two main types of growth against RH: monotonic and deliquescent. In the deliquescent

case, particles begin with absorbing water and grow only after the deliquescence RH (DRH) is reached. This DRH corresponds to the equilibrium RH above a saturated aqueous solution of the solute. Once RH surpasses the DRH, the aerosol undergoes a rapid increase in size and their refractive index tends toward that of liquid water, due to water uptake. If RH subsequently decreases, the droplet can remain in a metastable liquid state, supersaturated with respect to the solute, until it reaches a lower crystallization RH (CRH), at which point the particle recrystallizes. These phase transitions explain the pronounced hysteresis

observed in humidograms, which plot any hygroscopic property as a function of RH (Randriamiarisoa et al., 2006; Raut and Chazette, 2008b). The lower branch of the humidogram corresponds to particle humidification, while the upper branch corresponds to dehydration. These two branches intersect near the deliquescence and crystallization RH values, respectively, with *CRH < DRH*.

The growth $f_{scatt}(RH)$ of aerosol light scattering as a function of increasing RH is written as follows:



$$f_{scatt}(RH) = \frac{\sigma_{scatt}(RH)}{\sigma_{scatt,d}} \tag{12}$$

Where $\sigma_{scatt,d}$ is the scattering cross-section for dry aerosols, i.e. at $RH < 30\%$. In practice, RH is often taken at a reference value ($RH_{ref}$) at which the aerosol begins to grow, corresponding roughly to the DRH point, typically between 40 and 60% depending on the chemical compounds (Randriamiarisoa et al., 2006). Hänel (1976) proposed a parameterisation $\mathcal{H}_{scatt}$ which assumes that the increase in aerosol size as a function of RH is stable, with no abrupt change. This parameterisation, applied

to rising RH phases of the cycle, takes the form:

$$\mathcal{H}_{scatt}(RH) = (1 - RH)^{-\gamma} \tag{13}$$

where $\gamma$ is the Hänel scattering growth coefficient. Introducing $RH_{ref}$ gives:

$$f_{scatt}(RH) = \left( \frac{1 - RH}{1 - RH_{ref}} \right)^{-\gamma} \tag{14}$$

This relationship has been employed to characterise the hygroscopicity of aerosols using lidars (Bedoya-Velásquez et al., 2019;
Granados-Muñoz et al., 2015; Miri et al., 2024; Navas-Guzmán et al., 2019; Sicard et al., 2022) with the aerosol backscatter coefficient $\beta_a$ according to the relationship:

$$f_{\beta_a}(RH) = \frac{\beta_a(RH)}{\beta_a(RH_{ref})} = \left( \frac{1 - RH}{1 - RH_{ref}} \right)^{-\gamma} \tag{15}$$

Note that Hänel also defined the aerosol size growth factor, which is expressed as the ratio between the radius $r$ in the wet and quasi-dry conditions:

$$f_r(RH) = \frac{r(RH)}{r(RH_{ref})} \tag{16}$$

as following a similar law than the one of Eq. 14:

$$f_r(RH) = \left( \frac{1 - RH}{1 - RH_{ref}} \right)^{-\varepsilon} \tag{17}$$

The Hänel size growth coefficient, represented by the symbol $\varepsilon$, can be used to model the behaviour of aerosols in chemical transport and radiative models.

Finally, Hänel (1976) also describes the evolution of the complex refractive index $n$ of aerosols as a function of RH by the function $f_n(RH)$ defined by:

$$f_n(RH) = \frac{n(RH) - n_{H_2O}}{n(RH_{ref}) - n_{H_2O}} = \left( \frac{r(RH)}{r(RH_{ref})} \right)^{-3\varepsilon} \tag{18}$$

where $n_{H_2O}$ is the refractive index of pure water (1.35 at 355 nm). Eq. 17 and Eq. 18 will be included in a Mie code, as in Flamant et al. (1998) and Chazette et al. (2001)s, to verify the consistency between Mie scattering results on hydrated aerosols
and lidar measurements.

### 4    Results: aerosol hygroscopicity derived from Raman lidar

This section presents several case studies in which the atmosphere is suitable for the study of the hygroscopic properties of atmospheric aerosols trapped in the lower troposphere by Raman lidar. They are summarized in Table 2.

**Table 2. Cases studies and the associated i) CAMS-derived aerosol complex refractive index (ACRI), ii) Hänel exponent $\gamma$ assessed**
**for the lidar-derived aerosol backscatter coefficient ($\beta_a$) with the RH range considered, iii) $\varepsilon$ coefficient obtained using the Mie**





model, based on Mie theory, iv) modal radius of the accumulation mode, and v) $\gamma$ exponents of the aerosol extinction ($\alpha_e$) and scattering ($\sigma_s$) cross-sections using the Mie model.

| Case study | ACRI (CAMS) | Lidar derived $\gamma$ with its uncertainty (RH range (%)) | $\varepsilon$ (Mie simulation) | Modal radius (µm) (Mie simulation) | $\gamma$ using $\sigma_e$ and $\sigma_s$ (Mie simulation) |
|---|---|---|---|---|---|
| **Case study 1** <br> **25 May 2020** <br> **20:30–22:00 UTC** <br> **(0.7 – 1.7 km)** | 1.52 – 0.03.i | 0.38 ± 0.08 *(45 - 80 %)* | 0.20 | 0.105 | 0.43 and 0.47 |
| **Case study 2** <br> **26 May 2020** <br> **19:00–22:00 UTC** <br> **(0.7 – 1.7 km)** | 1.54 – 0.008.i | 0.50 ± 0.04 *(40 - 75 %)* | 0.32 | 0.085 | 0.73 and 0.75 |
| **Case study 3** <br> **27 May 2020** <br> **19:00–20:30 UTC** <br> **(0.7 – 1.7 km)** | 1.54 – 0.02.i | 0.75 ± 0.05 *(45 - 80 %)* | 0.37 | 0.080 | 0.87 and 0.92 |
| **Case study 4** <br> **31 May 2020** <br> **20:00–21:30 UTC** <br> **(1.3 – 1.8 km)** | 1.53 – 0.02.i | 0.31 ± 0.13 *(55 - 75 %)* | 0.19 | 0.100 | 0.40 and 0.43 |
| **Case study 5** <br> **2 June 2020** <br> **19:30–22:00 UTC** <br> **(1 – 2 km)** | 1.53 – 0.01.i | 0.53 ± 0.03 *(40 - 75 %)* | 0.31 | 0.085 | 0.72 and 0.74 |
| **Case study 6** <br> **25 May 2020** <br> **16:15–18:00 UTC** <br> **(1 – 1.9 km)** | 1.52 – 0.02.i | 0.30 ± 0.06 *(60 - 90 %)* | 0.18 | 0.100 | 0.39 and 0.42 |
| **Case study 7** <br> **25 August 2024** <br> **19:30–21:00 UTC** <br> **(1 – 1.6 km)** | 1.49 – 0.0008.i | 0.87 ± 0.15 *(65 – 85 %)* | 0.45 | 0.08 | 1.1 and 1.13 |
| **Case study 8** <br> **26 August 2024** <br> **18:00–20:00 UTC** <br> **(0.7 – 1.6 km)** | 1.50 – 0.006.i | 1.52 ± 0.16 *(60 - 90 %)* | - | - | - |



### 4.1 Case studies

The specific dates and times corresponding to each case study are indicated in Table 2. Eight hygroscopic growth cases were
identified in well-mixed layers. Across all case studies, atmospheric stratification was relatively similar in the aerosol layers
of interest. As shown in Figure 3, RH typically ranged from 40% to 80–90% within these layers. The deliquescence point is
assumed to be below the minimum RH found at the bottom of the aerosol layers, which we take as $RH_{ref}$, so that $\beta_a$ shows
monotonous growth from this point to the top of the aerosol layer. The WVMR remained nearly constant at around 6–7 g.kg$^{-1}$
$^{1}$ (9 g.kg$^{-1}$ for case study 8), and the potential temperature profiles indicated minimal vertical gradients, implying a well-mixed
and stable atmosphere.

The humidograms $f_{\beta_a}$ for the 8 hydroscopic cases are given in Figure 4, where $\beta_a$ is normalised by its value at $RH_{ref}$. The
corresponding Hänel law is represented by the black curve, for which the $\gamma$ coefficient is given in column 3 of Table 2. The
variability of $\gamma$, which is between 0.3 and 1.52, highlights the significant influence of atmospheric aerosol chemical
composition on hygroscopic growth, a point further developed in Section 6 thanks to CAMS chemical aerosol analysis and
reanalysis data. Although cases 4 exhibits a lower $\gamma$ value this does not necessarily imply lower aerosol hygroscopicity, given
that the range of relative humidity (RH) is limited. Stronger hygroscopic growth could have occurred, particularly in the
presence of highly hygroscopic compounds such as nitrate or sulphate salts, where DRH can be higher than 80% (e.g.
Randriamiarisoa et al., 2006, Table 1).

ERA5 data show that the first 5 case studies in May–June occurred under broadly similar synoptic conditions with the presence
of a persistent high-pressure system over Western Europe that gradually shifted eastward between 25 May and 2 June 2020.
This evolution explains the distinct air mass back trajectories shown in Figure 5a-e, where the three most representative
trajectories out of the 27 in ensemble mode are depicted. For case studies 1 to 3, air masses originated over the British Isles
before moving over the Benelux region and northern France, ultimately reaching Paris. In cases 4 and 5, the eastward shift of
the high-pressure system induced a more continental pathway, with air masses transiting from eastern Europe through Germany
before arriving over the Paris region.

The CAMS-derived aerosol optical thickness (AOT) reflects differences in aerosol loadings across the cases. On 25, 26 and
27 May, corresponding to cases 1, 2 and 3, AOT values at 550 nm over Paris were ~ 0.2, indicative of background pollution
levels (Chazette and Royer, 2017). Conditions became more stable during case 4 (AOT ~ 0.1), followed by an increase for
case 5 (AOT ~ 0.25). Note that elevated AOTs are observed over key European industrial regions, such as Benelux, the British
Isles, and the Ruhr area, through which the back trajectories pass. Therefore, the variations in AOT over Paris were probably
related to the long-distance transport of aerosols of different ages. This appears to be closely related to the hygroscopic growth
observed, as quantified by the $\gamma$ values given in Table 2. In contrast, the air masses for case study 6 originated from the west
due to a low-pressure system in the North Atlantic, coupled with the Azores High. Figure 5f shows low aerosol loading over
the Paris region (AOT ~ 0.15) with the arrival of clean marine air masses.

In August 2024, case studies 7 and 8 show air masses coming from the west, which also brought Atlantic air masses. This
circulation was caused by the presence of a low-pressure system in the northern part of the eastern Atlantic and the Azores
high. On 26 August, a high-pressure system settled over south-western France, maintaining the flow of air masses from the
west. Figure 5g and h show high AOT values over the Atlantic and over land for 26 August, suggesting the transport of marine
aerosols. Compared to other cases, a slight increase of the aerosol content is assessed over Paris (AOT ~0.2–0.3). The data
from the AERONET station in Paris corroborates this, with similar AOT recorded for an Angstrom exponent ranging from 1.1
to 1.4. Such values indicate the presence of a mixture of pollution aerosols and other types of particles.





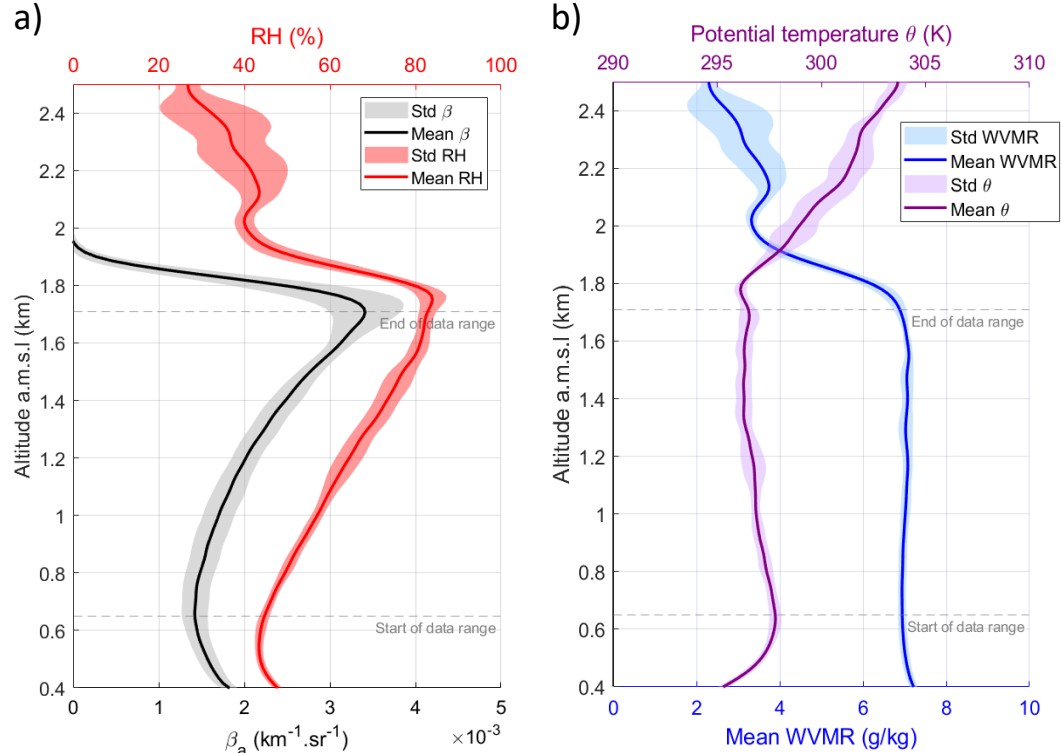

**Figure 3. Atmospheric parameters for case study 3 as a function of the altitude a.m.s.l.: a) the vertical profile of the backscattering coefficient in black and RH in red, b) the Water Vapour Mixing Ratio (WVMR) in blue, the potential temperature (Θ) in purple. The shaded areas represent the standard deviation of the data considered. Grey dotted lines give the altitude range considered. Case study 3 serves as a reference in terms of atmospheric conditions for the study of hygroscopicity. All the case studies present similar atmospheric conditions.**




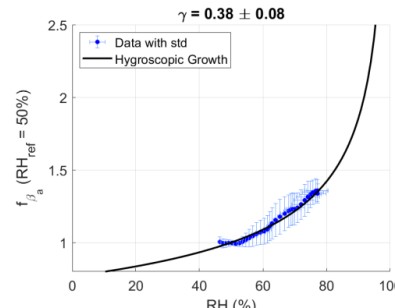
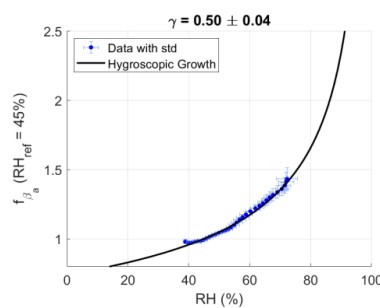

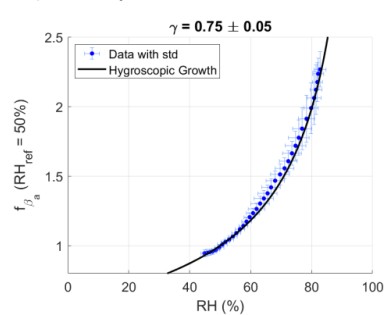
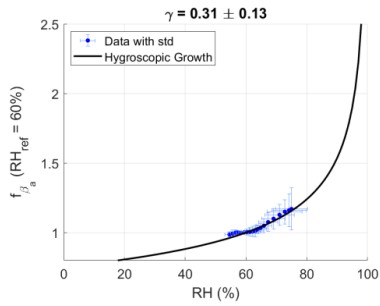

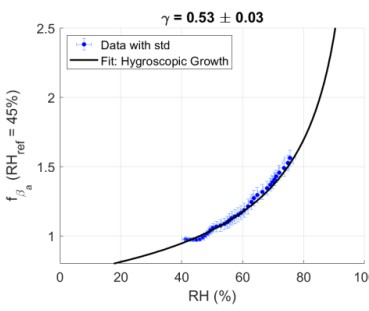
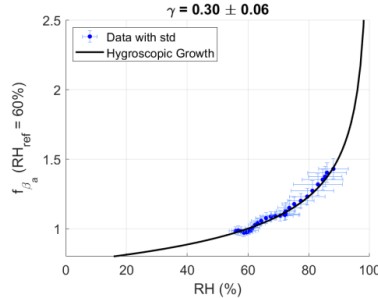

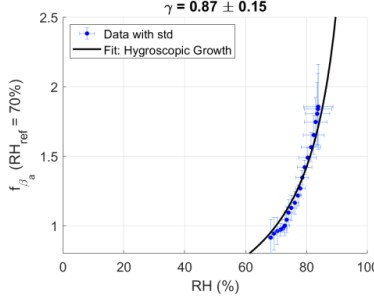
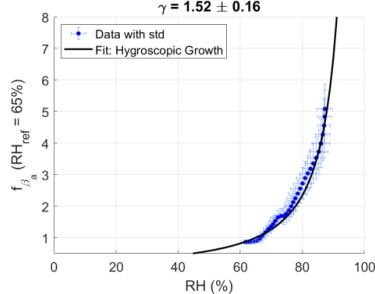


**Figure 4. Evolution of the backscattering function, defined as the ratio of the aerosol backscatter coefficient ($\beta_a$) normalised by the value of this coefficient for a reference relative humidity ($RH_{ref}$), as a function of RH for the 8 case studies. $RH_{ref}$ is taken as the mimimum observed RH. Above each figure is indicated the case study number and its date. The corresponding Hänel's law is**



represented by the black curve with the γ coefficient given above the figure. The blue points represent the lidar measurements, and the standard deviations are represented by the blue error bars. The temporal resolution of the profiles is 15 min and the vertical resolution is 15 m.

**Figure 5. Five-days back trajectories for the 8 hygroscopic case studies. The 3 lines represent 3 of the HYSPLIT ensemble mode whose arrival point is set in the time interval of each case study and at the middle of the studied aerosol layer. Each coloured dot**
**corresponds to the Aerosol Optical Thickness (AOT) given by CAMS reanalysis on the date corresponding to the HYSPLIT points. Each point on each trajectory is 3 hours apart. The letters 'H' and 'L' indicate the locations of high and low systems at 850 hPa, respectively.**



### 4.2 Uncertainties

The uncertainties on $\beta_a$ arise from to the molecular and aerosol contributions on $\beta$, the calibration coefficient $K$, and random
noise in the lidar signal. The uncertainty due to molecular scattering is reduced when the signal is corrected for molecular
transmission, making it negligible compared to other sources. Using ERA5 and the empirical relationship of Nicolet (1984),
the error on $\beta_m$ is assessed to be less than 1% (Chazette et al., 2014b). The signal-to-noise ratio (SNR) exceeds 100 in all
detection channels, resulting in random noise-induced uncertainties below 0.5%. The uncertainty due to $K$ is ~1 % calculated
using the variability in the molecular part of the profiles. The contribution of the aerosol transmission to the uncertainty is less
than 2%, as shown in Chazette and Totems (2023) and Laly et al. (2024). Assuming that each variable is independent, the total
relative uncertainty on $\beta_a$ is 2.5% for the entire altitude range considered.

Because of the nonlinear form of the expression for relative humidity (*RH*) (Eqs. 7 and 8), we derive the error on *RH* using a
Monte Carlo method as described in Royer et al. (2011) performing 1000 statistical realizations using Eq. 8. The Monte-Carlo
simulation is performed on each sample of *RH* generated by extracting values of $T$, *WVMR* and $P$ from actual WALI
measurements. Considering noise to be normally distributed, the respective standard deviation $\sigma$ applied to each parameter are
as follows: $\sigma_T \sim 0.5\ K$ for temperature (Baron et al., 2022; Totems et al., 2021); $\sigma_{WVMR} \sim 0.05\ g.kg^{-1}$ for the WVMR (Laly
et al., 2024; Totems et al., 2021); and $\sigma_P \sim 0.5\ hPa$ for pressure (Hersbach et al., 2020). Uncertainties are calculated at each
altitude with mean pressure, temperature and and lead to a standard deviation $\sigma_{RH} \sim 2.5$ % for RH at any altitude.

Similarly, the error on the Hänel coefficient for each case study is also determined with a Monte-Carlo approach but using Eq.
15. For each *RH* and $\beta_a$ lidar profile, statistical noise realizations are also generated by applying Gaussian noise to *RH* and $\beta_a$
with the standard deviations calculated before. The statistical uncertainties reported in Table 2 have been assessed by
calculating $\gamma$ each time.

## 5   The Hänel coefficient as derived from lidar measurements and Mie scattering

Above, the coefficient $\gamma$ has been estimated from Raman lidar measurements using the backscattering coefficient. While this
parameter can be estimated from lidar profiles under certain atmospheric conditions, it is not necessarily relevant for modelling
when evaluating aerosol growth (see Eq. 17) or changes to their complex refractive index (see Eq. 18) in wet conditions. This
section demonstrates how to relate the parameters $\gamma$ (Eq. 15) and $\varepsilon$ (Eq. 17) using the Mie scattering theory.

### 5.1 Hypothesis

Using Eqs. 17 and 18 in a Mie scattering model assuming spherical aerosols (Wriedt, 2012), it is possible to relate $\gamma$ and $\varepsilon$
provided that the size distribution of the aerosol particles is known. Previous work by Randriamiarisoa et al. (2006) showed
that the accumulation mode is the main contributor to aerosol scattering in the Paris region, as well as being the most sensitive
to relative humidity (RH). In the case studies, PDR values of less than 4% were observed, indicating a negligible contribution
from dust-like aerosols. Therefore, it can be assumed that the coarse mode has a lesser impact on optical properties than the
accumulation mode. It should also be noted that the nucleation mode of the size number distribution, composed of particles
with a modal radius of ~ 30 nm, accounts for less than 4% of total aerosol scattering (Randriamiarisoa et al., 2006). In all
observations in the Paris region, the accumulation mode followed a log-normal size number distribution with a modal radius
between 80 and 120 nm, and a geometrical dispersion of about 1.5 (Randriamiarisoa et al., 2006; Raut and Chazette, 2009).
This type of distribution remains consistent throughout the mixing layer and changes very little, as demonstrated by Chazette
et al. (2005). Furthermore, data from the photometer at the AERONET station in Palaiseau (https://aeronet.gsfc.nasa.gov/, last
access 7 July 2025) provides information on the integrated volume distributions of aerosols (Figure 6a). These volume
distributions can then be used to derive the normalised number size distribution. (Figure 6b). Fitting a log-normal distribution
to the particle size distribution derived from AERONET measurements yields a modal radius of 0.08 µm and a geometric




dispersion of 1.48, which is consistent with previous findings. Hereafter, the dispersion will be fixed, and the modal radius will be determined alongside $\varepsilon$. Once the geometric dispersion has been fixed at 1.5, the dry modal radius that best fits the

power law (Eq. 15) is derived. The value of $\varepsilon$ that most closely matches the measured value of $\gamma$ is then chosen as the solution. This calculation requires an estimate of the complex refractive indices of aerosols (ACRIs). This can be achieved by considering the chemical speciation provided by CAMS reanalyses/analyses alongside the known ACRIs of the individual chemical compounds in the aerosols. These are provided in Table 3, along with the densities of the chemical compounds and the associated bibliographic references. The real parts of ACRIs are close to each other in the UV/visible spectrum, except for

black carbon and sulphate compounds, which therefore cause variations in the values from one case study to another. The imaginary part of the ACRI is mostly influenced by black carbon.

Assuming that the particles are homogeneously mixed and that each compound contributes to scattering and absorption in proportion to its volume fraction, the total ACRI ($n_{tot}$) of a mixture can be expressed as follows (Tombette et al., 2008):

$$n_{tot} = \frac{\sum_i n_i . V_i}{\sum_i V_i} \tag{21}$$

with $n_i$ the ACRI of component $i$ and $V_i$ its volume in m³. From the mixing ratios ($r_i$ in kg.kg⁻¹) obtained with CAMS and the density of each component ($\rho_i$ in kg.m⁻³), we obtain the following relationship:

$$n_{tot} = \frac{\sum_i n_i . \frac{r_i}{\rho_i}}{\sum_i \frac{r_i}{\rho_i}} \tag{22}$$

The ACRI for each case study, considering only the particles in the accumulation mode, are given in Table 3.

For the Mie simulations, it is therefore assumed that a monomodal log-normal distribution with a geometrical dispersion of

1.5 is applicable with the CAMS-derived ACRI. Using the power laws established by Hänel (1976), the pair ($r_{mod}$, $\varepsilon$) that best fits the measured humidogram in Figure 4 is sought.

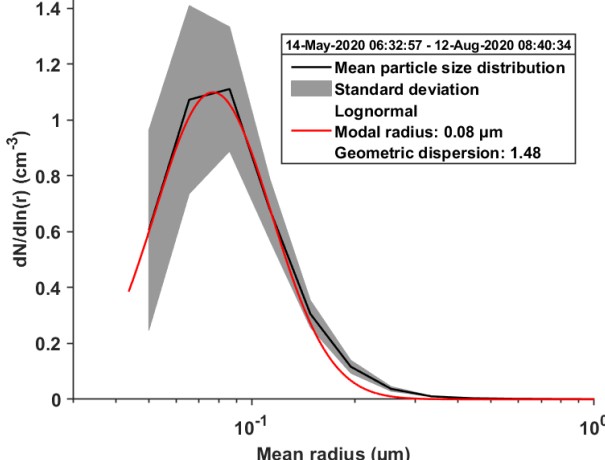

**Figure 6. Normalised number size distribution derived from AERONET (black curve) and the log-normal size distribution adjusted (red curve). The grey shaded area represents data variability.**

**Table 3. Table showing the properties of different types of aerosols: their typical emission sources, their density and their complex refractive index (ACRI). POM stands for Particulate Organic Matter.**

| Chemical compound of the aerosol | Main sources | Density (g.cm⁻³) | ACRI (300 – 1000 nm) | References |
|---|---|---|---|---|




| | | | | |
|---|---|---|---|---|
| **Ammonia** | Agricultural emissions (fertilizers, livestock) | ~ 1.75 | 1.52 | (Seinfeld and Pandis, 2000) |
| **Sulfate** | Fossil fuel combustion (coal, oil) | ~ 1.75 | 1.43 | (Seinfeld and Pandis, 2000) |
| **Nitrate** | Vehicle emissions ($NO_x$), biomass burning | ~ 1.75 | 1.55 | (Seinfeld and Pandis, 2000) |
| **Dust** | Desert regions, natural erosion | 2.3 - 2.6 | 1.50 – 0.01.i | (Chazette and Liousse, 2001; Seinfeld and Pandis, 2000) |
| **Black carbon** | Biomass burning, residential heating, forest fires | ~ 1.5 | 1.95 – 0.7.i | (Chazette and Liousse, 2001; Seinfeld and Pandis, 2000) |
| **POM** | Combustion (fossil fuels, biomass), plant debris | ~ 1.5 | 1.55 – 0.005.i | (Chazette and Liousse, 2001) |
| **Sea salt** | Sea aerosols | ~ 2.1 | 1.50 | (Chazette and Liousse, 2001) |
| **Water soluble** | Mixture of pollution and natural aerosols | ~ 1.7 | 1.53 – 0.005.i | (Chazette and Liousse, 2001) |

## 5.2 Results derived from Mie theory

Figure 7 provides an example of the adjustment of the $\gamma$ coefficients for the effective cross sections of backscattering ($\sigma_r$),

extinction ($\sigma_e$), and scattering ($\sigma_s$). As shown Table 2, the values of $\varepsilon$ range from 0.19 to 0.45 for modal radii close to the





expected range of ~ 0.08–0.1 µm. Chazette and Liousse (2001) report $\varepsilon$ values of 0.26 for aerosols mainly due to traffic in the Thessaloniki city in Greece. Such value is close to the one derived by Hänel (1976) for water soluble aerosols where $\varepsilon = 0.285$. Similar values have been retrieved in Paris area by Randriamiarisoa et al. (2006) and Raut and Chazette (2008b) with values of 0.26 and 0.29, respectively. It should be noted that these data were collected over short observation periods. Therefore, they

cannot account for all the variability in aerosol chemical composition, which can change significantly from one case to the next. This variability can also be significant within a single day (Randriamiarisoa et al., 2006).

Using a Mie model also provides an opportunity to evaluate the effective extinction and scattering cross-sections and derive the corresponding $\gamma$ exponents. These are also reported in Table 2. The $\gamma$ values derived from backscattering lidar measurements are significantly lower. This shows that working with backscattering or extinction is not equivalent, and that the lidar ratio (the

ratio of extinction to backscattering coefficients) also depends on RH. This difference is largely due to the increase in aerosol size, which favours forward scattering. It should also be noted that the $\gamma$ exponent is slightly higher via the scattering cross section than via the extinction cross section. Consequently, there is a slight dependence of the absorbing part of the aerosol against *RH*. This can be explained by the light diffractive effect due to the liquid film formed around the aerosol. Some authors neglect this contribution (e.g. Dawson et al., 2020).

It was not possible to determine the values of $\gamma$ and $\varepsilon$ for case study 8. This is due to the bimodal nature of the aerosol distribution. A second mode centred on a modal radius of ~0.14 µm interferes with the accumulation mode centred on ~0.1 µm as highlighted from AERONET data. This second mode is most likely to be composed mainly of sea salt. Therefore, it is likely to be highly hydrophilic.

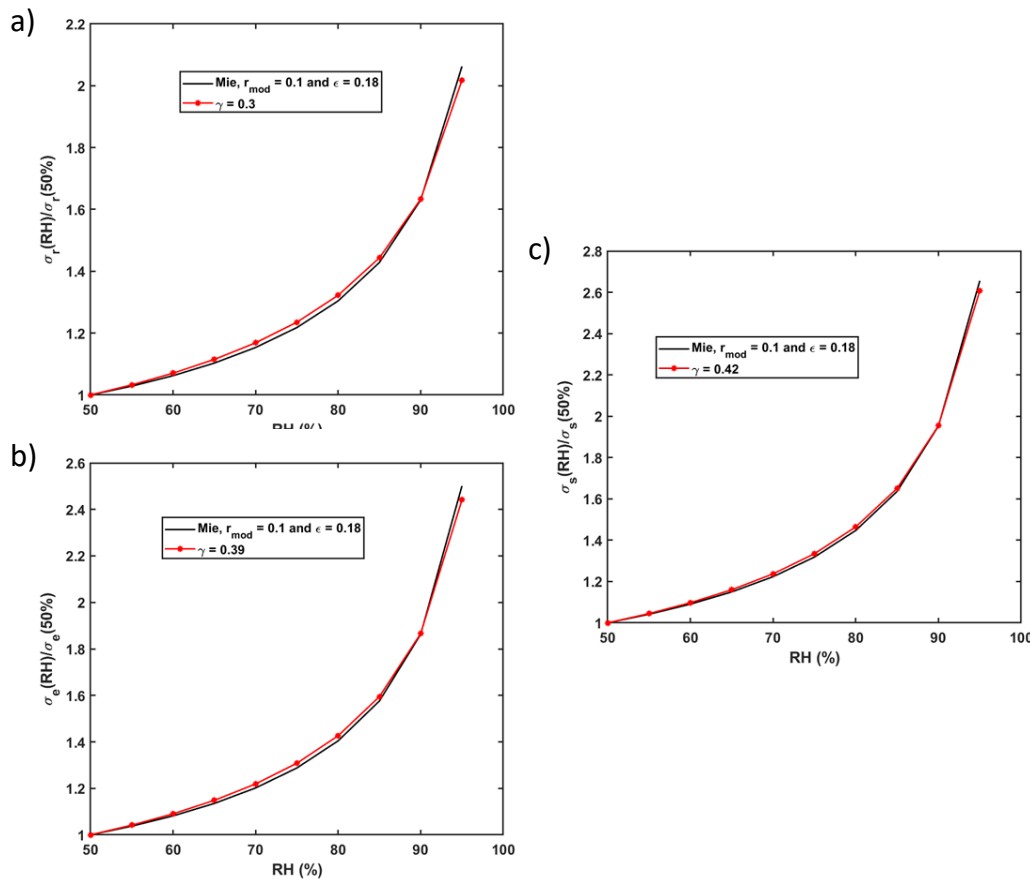



**Figure 7. Humidograms for Case Study 1 on 25 May 2020. a) The aerosol backscattering coefficient humidogram, derived from Raman lidar measurements, is shown in red and superimposed on the theoretical humidogram derived from Mie scattering, shown in black. b) [c)] The aerosol extinction [scattering] coefficient humidogram, derived from both the Raman lidar measurements and Mie scattering, is shown in red and superimposed on the theoretical humidogram derived from Mie scattering, shown in black. The calculation was performed using the same ε.**

**6    Evidence of aerosol hygroscopicity: coherence between CAMS and lidar**

To investigate aerosol hygroscopicity, CAMS analysis and reanalysis data are used to track the temporal evolution of key aerosol chemical compounds within the atmospheric column. CAMS analysis provides mass mixing ratios for nitrate ion ($NO_3^-$), ammonium ion ($NH_4^+$), sulfate ion ($SO_4^{2-}$), and sea salt components, while reanalysis data include dust, as well as hydrophilic and hydrophobic fractions of organic and black carbon components. Figure 8 illustrates the relative contributions

and mixing ratios of the chemical compounds resulting from the synergy of the two datasets in each case study.

Cases 1 to 5 have been selected within late May to early June 2020. During this period, the aerosol properties evolved under similar synoptic conditions. Although CAMS provides valuable information about the chemical composition of aerosols, it does not give explicit information on the state of aerosols, such as their ageing. However, trends observed in aerosol content and spatial distributions can be used in conjunction with atmospheric transport to formulate hypotheses about potential aerosol

ageing. Cases 6 and 7, occurring at different times, are considered separately as points of comparison.

Figure 8 shows that hydrophilic organics, nitrates, sulfates, and ammonium dominate the aerosol burden in the first six case studies, with both their proportions and mixing ratios varying across cases. In cases 2 and 4, hydrophilic organics account for about 25-30% of the aerosol composition, while in cases 3 and 5, their proportion increases to 35-40%. These compositional differences align with lidar-derived γ values: case 3 exhibits the highest γ (0.75) of the six first case studies, coinciding with

the largest relative contribution of hydrophilic organics and elevated $NO_3^-$ mixing ratios. Case 2 shows a γ of 0.50, consistent with an overall increase in hygroscopic species, particularly for $NO_3^-$ and $NH_4^+$, compared to case 1. In case 5, γ remains high (0.53), despite slightly lower $NO_3^-$ mixing ratio, likely due to a compensating increase in $SO_4^{2-}$ content. This suggests the possible formation of ammonium nitrate and ammonium sulphate, both highly hygroscopic (Seinfeld and Pandis, 2000; Tang, 1996). In contrast, case 1 displays a lower γ (0.38), despite a notable fraction of hygroscopic species. A key difference is the

much lower total aerosol mixing ratio (~ 10 µg.kg⁻¹ compared to ~ 20 µg.kg⁻¹ for cases 2, 3 and 5), suggesting that the aerosols were relatively young and not related to remote transport. Similarly, case 4, with the lowest γ (0.31), shows reduced proportions of hydrophilic organics and the lowest aerosol load (< 10 µg.kg⁻¹). These cases highlight the combined importance of both chemical composition and total aerosol mixing ratio in modulating hygroscopic behaviour.

Figure 5a-e further support this interpretation by showing increased AOT during cases 2, 3, and 5, suggesting rather old

aerosols, which were probably transported from remote emission sources. These cases coincide with higher γ values and enriched hygroscopic composition and aerosol load, suggesting the influence of transported aerosols that have undergone atmospheric aging. Although CAMS does not explicitly model aerosol aging, the spatial AOT patterns and enhanced mixing ratios suggest that these aerosols originated from distant emission regions, such as the Benelux area, and likely experienced oxidative processing during transport. This is consistent with known atmospheric mechanisms, in which aging increases the

solubility and hygroscopicity of aerosols, particularly through oxidation of organic matter (Jimenez et al., 2009). However, it should be noted that reanalyses are given every 3 hours and at a horizontal resolution of 0.75°x0.75°. The points plotted in Figure 5 do not allow us to determine what happened locally at the lidar site.

Case 6 (4 August 2020) presents a different atmospheric situation. Although it exhibits aerosol proportions and mixing ratios similar to case 1, with less sea salts, its γ value is slightly lower (0.30 vs. 0.38). The hygroscopic behaviour observed is also

comparable to that of case 4, which showed the lowest γ value (0.30) among the first five cases. Like case 4, case 6 is characterized by a lower fraction of hydrophilic organics and moderate levels of nitrates and sulfates. These similarities suggest the presence of less hygroscopic aerosol types that have undergone limited atmospheric ageing or processing. This is consistent



with the assessment of the aerosol load (~ 10 µg.kg⁻¹) derived from CAMS. This interpretation aligns with the absence of strong transport features or enhanced AOT in the CAMS data for that day.

Case studies 7 and 8, on 25 and 26 August 2024, exhibit the highest $\gamma$ coefficients ($\gamma = 0.87$ for case study 7 and $\gamma = 1.52$ for case study 8). In Figure 8, aerosol type was dominated by marine aerosols (~ 60 % of the aerosol load for case study 7 and ~ 40 % of the aerosol load for case study 8), which have very high hygroscopicity properties (Haarig et al., 2017; Randriamiarisoa et al., 2006). As can be seen with the back trajectories (Figure 5g), these aerosols were likely transported from the Atlantic Ocean. The marine aerosols were mixed with pollution-type aerosols, as suggested by the significant presence of nitrate, sulphate and ammonium coupled with hydrophilic organics. However, in case 7, the relative concentration of nitrate, sulfate,

ammonium and organic compounds was lower than in case 8, indicating purer marine aerosols. The CAMS results for these cases are thus consistent with the Hänel coefficient obtained by lidar, indicating aerosols with very high hygroscopic properties.

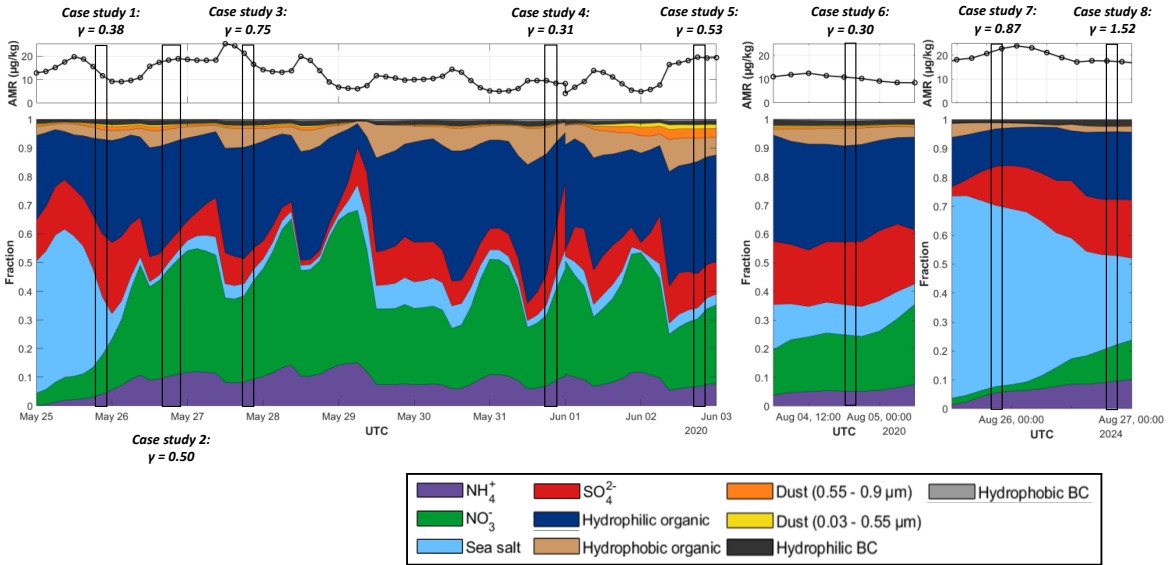

**Figure 8. Temporal evolution of the fraction of different atmospheric compounds. Above is the temporal evolution of the aerosol**
**mixing ratio (AMR in µg.kg⁻¹) with all species combined. The black rectangles indicate the time period of each case study with the associated Hänel ($\gamma$) coefficient based on the humidogram of the lidar aerosol backscatter coefficient. Colours represent the different aerosol chemical compounds as indicated in the legend. For these figures, the outputs of the analysis and reanalysis have been combined, with the chemical content extracted between 950 and 850 hPa, including the relevant aerosol layers.**

## 7    Conclusion

This study demonstrates the capability of ground-based Raman lidars, such as the WALI instrument, to retrieve information on aerosol hygroscopic properties in the atmospheric column. By applying Hänel's theory to lidar-derived backscatter coefficients, we were able to quantify hygroscopic growth through the $\gamma$ coefficient for 8 case studies observed between May and August 2020 in Saclay (France) and in August 2024 in Paris. To better understand the origin and nature of the aerosols responsible for these hygroscopic signatures, we used data from CAMS analyses and reanalyses. CAMS outputs provide a

temporal and chemical perspective on aerosol composition, including mixing ratios of key hygroscopic species such as nitrate, ammonium, sulphate, sea salt, and organic components. Literature values for the hygroscopicity of these species were used as a reference to interpret the CAMS-derived composition. This reference served to frame the comparison with the Hänel coefficients retrieved from lidar measurements, for which a strong consistency in temporal evolution was observed. Meteorological context and AOT maps further supported the interpretation of long-range transport and its potential influence

on aerosol aging and solubility. While this approach offers valuable insight, certain limitations must be acknowledged. CAMS data, though chemically resolved, do not account for internal aerosol processes such as oxidation, phase transitions, or





temperature-dependent solubility changes, all of which can affect hygroscopicity. For instance, the temperature-dependent solubility of organic aerosols, as highlighted by Jaffrezo et al. (2005), may lead to underestimations in modelled hygroscopic growth under warm conditions. Furthermore, differences in vertical resolution and chemical apportionment in CAMS analysis

and reanalysis may introduce uncertainties when comparing to lidar retrievals. To complement the observational analysis, the relationship between the hygroscopicity parameter $\gamma$ and the Hänel size growth coefficient $\varepsilon$ has been studied using Mie theory. With the hypothesis of a monomodal accumulation mode, a wide range of $\varepsilon$ values between 0.19 and 0.45 was derived from the coupling of lidar-derived backscatter coefficient of aerosols, Mie theory and the CAMS-derived ACRI. Such values can be compared to existing values in the literature, ranging from 0.26 to 0.29 given for pollution aerosols. The higher and lower

values of $\varepsilon$ obtained by Mie simulation are interesting to compare because they correspond to a mixture of aerosols that are either less or more hygroscopic, consistent with CAMS data. Coupling the Mie and Hänel models also reveals significant differences in behaviour with respect to RH for the extinction and backscatter coefficients.

Overall, the $\gamma$ coefficients retrieved in this work are consistent with those found in the literature (as summarized in Table 4), lending confidence to the methodology. This work highlights the relevance of integrating lidar observations with chemical

transport models like CAMS to characterize aerosol-humidity interactions. Beyond the present results, systematic lidar measurements of hygroscopic growth could support air quality forecasting by improving PM mass estimates under varying humidity, enhance climate model estimates of aerosol radiative forcing through vertically resolved RH-dependent optical properties, and provide valuable input for model evaluation, data assimilation, and satellite product validation. Long-term multi-instrument observations could also offer opportunities to classify regional aerosol regimes by hygroscopic behaviour

and to contribute to early warning systems for high-pollution events for instance.

**Table 4. Characteristic table of the hygroscopic power of different types of aerosols given with the Hänel coefficient ($\gamma$) at 355 nm and 532 nm, as retrieved from the aerosol backscattering coefficient data obtained from lidars. The various reference works are given in the first column. The results of this article are given in the last line. Organics[1] indicates a greater amount of organics compared to Organics[2].**

| Works | Aerosol type | $\gamma$ (355nm) | $\gamma$ (532nm) |
|---|---|---|---|
| Haarig et al. (2017) | Sea salts | 1.08 | 1.49 |
| Granados-Munoz et al. (2015) | Sea salts – sulfates<br>Regional pollution – mineral dust | -<br>- | 1.1<br>0.56 |
| Fernandez et al. (2015) | Sea salts- nitrates – organics<br>Nitrates - organics | -<br>- | 0.88<br>0.59 |
| Sicard et al. (2022) | Local pollution – sea salts<br>Regional pollution – sea salts | 0.81<br>0.71 | 0.73<br>0.70 |
| Chen et al. (2019) | Sulfates – nitrates – organics<br>Sulfates – organics (clean) | -<br>0.18 | 0.65<br>- |



| | | | |
|---|---|---|---|
| **Bedoya – Velazquez et al. (2018)** | Biomass burning – regional pollution | 0.40 | 0.48 |
| **Navas – Guzman et al. (2019)** | Biomass burning – rural<br>Mineral dust | 0.48<br>0.18 | -<br>- |
| **Perez-Ramirez et al. (2021)** | Sulfates – organics[1]<br>Sulfates – organics[2] | 0.46<br>0.65 | 0.39<br>0.38 |
| **Miri et al. (2024)** | Regional pollution<br>Biomass burning | -<br>- | 0.47<br>0.5 |
| **This study** | Case study 1: Regional pollution (Nitrates – Sulfates – Ammonium – Organics[1]) | 0.38 ± 0.08 | - |
| | Case study 2: Regional pollution (Nitrates – Sulfates – Ammonium – Organics[2]) | 0.50 ± 0.04 | |
| | Case study 3: Regional pollution (Nitrates – Sulfates – Ammonium – Organics[2]) | 0.75 ± 0.05 | |
| | Case study 4: Regional pollution (Nitrates – Sulfates – Ammonium – Organics[1]) | 0.31 ± 0.13 | |
| | Case study 5: Regional pollution (Nitrates – Sulfates – Ammonium – Organics[2]) | 0.53 ± 0.03 | |



| Case study 6: Regional pollution (Nitrates – Sulfates — Organics[1]) | 0.30 ± 0.06 | |
| Case study 7: Sea salts and regional pollution (Nitrates – Sulfates — Organics[1]) | 0.87 ± 0.15 | |
| Case study 8 : Sea salts and regional pollution (Nitrates – Sulfates — Organics[1]) | 1.52 ± 0.16 | |


**Data availability.** All data sets have been downloaded from the respective model websites.

**Author contributions.** FL wrote the paper, processed and analyse the data, performed the calculations and contributed to the 2024 field campaign. PC coordinated the field campaigns, wrote the paper, processed the data, ran calculations and contributed to the proofreading of the paper. JT prepared the lidar, took part in the field campaigns, performed preliminary processing and analysis of the data, and helped proofreading the paper. VC contributed to the analysis of chemical pollution species and to the proofreading of the paper. AM and DL contributed to the proofreading of the paper and helped with the field campaign in 2024.

**Competing interests.** The authors declare that they have no conflict of interest.

**Acknowledgements.** This work has been supported by the Institut National des Sciences de l'Univers (INSU) of Centre National de la Recherche Scientifique (CNRS) and by the Direction de la Recherche Fondamentale (DRF) of Commissariat à l'Energie Atomique et aux Énergies Alternatives (CEA). Thanks are extended to the Observatoire de Paris for hosting the field experiment campaign in the summer of 2024.

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
