# Peer review of "How does humidity affect lidar-derived aerosol optical properties, and how do they compare with CAMS?"

_EGUsphere, 2025_

## Referee Comment (RC1)

This manuscript investigates aerosol hygroscopic growth within the well-mixed planetary boundary layer, deriving backscatter coefficients as a function of relative humidity. The authors relate these measured growth factors to the aerosol chemical composition from the CAMS model. The study is executed at a high technical level by established experts in the field. While over half of the manuscript covers methodological details—including many well-known formulas—this comprehensive approach is justified as it provides a valuable, self-contained reference for the reader. I find this to be a high-quality contribution that matches the standards of AMT. I therefore recommend acceptance after the authors address the following minor comments:

Title. "…lidar-derived aerosol optical properties…". Actually only backscattering coefficient is presented.
It is a pity, that authors don't provide the lidar ratios. Dependence of lidar ratio on RH for different aerosols would be interesting.

Abstract. "…The results demonstrate the capability of Raman lidar to constrain aerosol hygroscopicity, offering valuable input to chemistry-transport models and helping to reduce uncertainties in climate projections related to aerosol-cloud interactions." This is very strong statement. I agree that analysis of backscattering at variable RH is an interesting approach to get information about aerosol mixture; still results presented are insufficient to access such goal.

Ln.217. Formulas for backscattering calculation were first published by Ansmann et al. 1992. Corresponding reference is needed.

Eq.16. This formula was used by Hanel for a single particle. When it is applied to aerosol with PSD, some assumptions are made. Should be discussed. The same is for Eq.18.

Section 4.4 Do authors consider change of the Angstrom exponent with RH?

Ln.435. Dependence of lidar ratio on RH was analyzed in recent paper

Fig.8. From this Fig. I conclude that it is not so simple to relate CAMS data with measured $\gamma$. For example Cases 3 and 4 have similar composition, but very different $\gamma$. As authors mention, aging can be also important. This is why I wrote above, that statement in Abstract is too strong.

---

## Referee Comment (RC2)

[revised manuscript text omitted]
     | _                     | 0.4°x0.4°/         | 0.25°x0.25°       | 0.25°x0.25° |
|----------------|-----------------------|--------------------|-------------------|-------------|
| resolution     |                       | 0.75°x0.75°        |                   |             |
| Vertical       | 15 m                  | 25 – 100 hPa       | -                 | -           |
| resolution /   |                       |                    |                   |             |
| Pressure level |                       |                    |                   |             |
| resolution     |                       |                    |                   |             |
| Altitude/Level | 0 - 3 km              | 0 – 4.5 km         | 850 hPa (~1.5 km) | -           |
| considered     |                       |                    |                   |             |
|                |                       |                    |                   |             |
| Temporal       | Native: 1 min         | 12 hours / 3 hours | 1 hour            | 1 hour      |
| resolution     | Operational: 15 min   |                    |                   |             |
| Altitude range |                       | -                  | -                 | -           |
| (a.g.l.)       |                       |                    |                   |             |
| Raman          | 0 – 2 km (Daytime)    |                    |                   |             |
| channels       | 0 – 10 km (Nighttime) |                    |                   |             |
|                |                       |                    |                   |             |
|                |                       |                    |                   |             |
| Elastic        | 0-10  km              |                    |                   |             |
| channels       |                       |                    |                   |             |
|                |                       |                    |                   |             |
|                |                       |                    |                   |             |
|                |                       |                    |                   |             |

[revised manuscript text omitted]

Mineral dust                                                                                                           | 0.48
0.18                    | -
-
-  |
| Perez-
Ramirez et
al. (2021)     | Sulfates – organics 1
Sulfates – organics 2                                                                              | 0.46
0.65                    | 0.39
0.38 |
| Miri et al. (2024)                     | Regional pollution Biomass burning                                                                                                                | -                               | 0.47
0.5  |

[revised manuscript text omitted]

---

## Author Comment (AC1)

**Responses to Review #2**

The authors would like to thank the reviewer for his valuable comments which helped improving the quality of the manuscript. Our point-by-point responses to the reviewer's comments appear in bold below.

**Abstract:**

Line 12: allow study → allow to study the

The correction has been made.

Line 12: water vapour → water vapour profiles

The correction has been made.

Line 13: lidar derived backscatter coefficients → lidar-derived aerosol backscatter coefficients

The correction has been made.

Line 15: long range pollution transport → long range air pollution transport

The correction has been made.

Line 22: of Raman lidar → of a Raman lidar

The correction has been made.

**Introduction:**

Line 27: The Earth surface → the Earth's surface

The correction has been made.

Line 28: The reference "IPCC, 2023" has been added.

Line 36: More recent works on CCN have been added ("Che et al., 2016; Paramonov et al., 2015"). We thank the reviewer for the last reference.

Line 56: and backscattering coefficient → and aerosol backscattering coefficient

The correction has been made.

Line 73: the origin of air masses → the origin of sampled air masses

The correction has been made.

Section 2:

Line 96: and thus play  $\rightarrow$  and thus, play

The correction has been made

Line 102: of various pollutants → of various air pollutants

The correction has been made.

Line 117: uses as emitter pulsed → uses as emitter a pulsed

The correction has been made.

Line 125: The following references on the identification of aerosols using depolarisation of the signal have been added: Chazette and Totems, 2023; Dieudonné et al., 2015.

Section 3:

Line 183: The following references on first Stokes vibrational lines have been: Whiteman et al., 1992; Weitkamp, 2005.

Line 232: Angstrom → Ångström

The correction has been made.

Line 235: References on low PDR for pollution particles have been added: Dieudonné et al., 2015; Mylonaki et al., 2021.

Line 266: Where → where

The correction has been made.

Section 4:

Line 353: Five-days back trajectories → Five-days air mass back trajectories

The correction has been made.

Line 474: Figure 5a-e further support  $\rightarrow$  Figures 5a-e further support

The correction has been made.

**Acknowledgements:**

Line 551: Thanks are extended to the  $\rightarrow$  We acknowledge the

The correction has been made.

---

## Author Comment (AC2)

**Responses to Review #1**

The authors would like to thank the reviewer for his valuable comments which helped improving the quality of the manuscript. Our point-by-point responses to the reviewer's comments appear in bold below.

Title. "...lidar-derived aerosol optical properties...". Actually only backscattering coefficient is presented.

It is a pity, that authors don't provide the lidar ratios. Dependence of lidar ratio on RH for different aerosols would be interesting.

Section 4.4 Do authors consider change of the Angstrom exponent with RH?

We thank the reviewer for these very relevant comments. Indeed, only the dependence of the backscattering coefficient on RH was considered in this study. The title is indeed not precise enough in this regard. It would be interesting to study the behaviour as a function of RH of all the optical parameters that can be calculated with lidar: Angström coefficient, backscattering coefficient (done here), LR and perhaps even depolarisation. We believe that these elements could constitute an entire article. The article under consideration here already presents many ideas, and a significant amount of work would be required to address the various points suggested, which are nevertheless very interesting. For this reason, we will not add these elements here.

Abstract. "...The results demonstrate the capability of Raman lidar to constrain aerosol hygroscopicity, offering valuable input to chemistry-transport models and helping to reduce uncertainties in climate projections related to aerosol-cloud interactions." This is very strong statement. I agree that analysis of backscattering at variable RH is an interesting approach to get information about aerosol mixture; still results presented are insufficient to access such goal.

Fig.8. From this Fig. I conclude that it is not so simple to relate CAMS data with measured  $\gamma$ . For example Cases 3 and 4 have similar composition, but very different  $\gamma$ . As authors mention, aging can be also important. This is why I wrote above, that statement in Abstract is too strong.

We thank the reviewer for this comment. Indeed, the sentence is too assertive. It has therefore been modified as follows in order to nuance the statement: "The results illustrate the potential of Raman lidar observations to provide valuable constraints on aerosol hygroscopicity, offering complementary information for chemistry-transport models and contributing to reducing uncertainties in aerosol-cloud interaction estimates."

Ln.217. Formulas for backscattering calculation were first published by Ansmann et al. 1992. Corresponding reference is needed.

The reference "Ansmann et al. 1992" has been added. We thank the reviewer.

Eq.16. This formula was used by Hanel for a single particle. When it is applied to aerosol with PSD, some assumptions are made. Should be discussed. The same is for Eq.18.

We thank the reviewer for this insightful comment. We agree that the Hänel law was originally derived for a single particle and that its application to an aerosol mixing involves implicit assumptions. In our study, we assume that the aerosol population is internally and homogeneously mixed, which allows the use of an effective hygroscopicity parameter to describe the ensemble optical response. This assumption is consistent with previous lidar-based studies applying the Hänel formulation to derive hygroscopic growth although these works did not explicitly discuss the mixing-state limitation. We have clarified this point in the revised manuscript (Section 3.5): "It is worth noting that the Hänel parameterization was originally developed for a single particle. Its application to aerosol ensembles with a particle size distribution therefore involves the implicit assumption that particles are internally and homogeneously mixed and exhibit a uniform hygroscopic response."

Ln.435. Dependence of lidar ratio on RH was analyzed in recent paper

We thank the reviewer for this comment, which has enabled us to deepen our knowledges on this point. After checking in the literature, we have added the following reference: "Zhao et al., 2017".